# Emotion Dynamics and Emotion Regulation in Anorexia Nervosa: A Systematic Review of Ecological Momentary Assessment Studies

**DOI:** 10.3390/ijerph192013659

**Published:** 2022-10-21

**Authors:** Magdalena Wayda-Zalewska, Piotr Grzegorzewski, Emilia Kot, Ewa Skimina, Philip S. Santangelo, Katarzyna Kucharska

**Affiliations:** 1Institute of Psychology, Cardinal Stefan Wyszyński University in Warsaw, 01938 Warsaw, Poland; 2Faculty of Psychology, University of Warsaw, 00183 Warsaw, Poland; 3Faculty of Psychology, University of Hagen, 58097 Hagen, Germany

**Keywords:** anorexia nervosa, ecological momentary assessment, experience sampling method, ambulatory assessment, emotion dynamics, emotion regulation

## Abstract

Altered emotion dynamics and emotion regulation (ER) have been indicated in theoretical descriptions of abnormal emotional functioning, which contributes to the development and maintenance of anorexia nervosa (AN). Ecological momentary assessment (EMA) has recently become popular in research on eating disorders. It is a source of new insights into the psychopathology of AN as it enables intensive long-term tracking of everyday experiences and behaviours of individuals through repeated self-reports. The following systematic review aims to synthesize research on the use of EMA when evaluating emotion dynamics and ER in AN. Specific studies were identified with the use of MEDLINE, PsycINFO, and Scopus databases. A supplemental search was performed in reference lists of the relevant publications. As a result, 27 publications were identified and included in the systematic review. The findings from the reviewed studies point to various disturbed components of emotion dynamics as well as to unique associations of maladaptive ER strategies with specific abnormalities in emotion dynamics in AN. Limitations of the studies were discussed as well. An outlook for further research in the field was provided in the last section of the paper.

## 1. Introduction

Anorexia nervosa (AN) is characterised by both mental and physical suffering. Its diagnostic symptoms include intense fear of weight gain, disturbances in body image and being underweight, accompanied by behaviours aimed at reducing energy intake (e.g., restricted eating) as well as increasing energy expenditure (e.g., excessive physical activity), and, in certain individuals, also by binge eating and purging episodes (i.e., self-induced vomiting, misuse of laxatives, diuretics or enemas [1]). Despite advances in clinical research on AN [2] and an increasing number of available evidence-based treatment approaches [3], the results of large-scale longitudinal studies show that remission occurs only in up to 40% of patients. However, although improved, the amount of consumed food and general pathology in this group do not reach the levels of healthy individuals [4]. There is also limited knowledge about the processes crucial to clinical improvement that should be targeted to reduce the level of eating pathology, which hampers the development of effective therapeutic interventions [5].

### 1.1. Emotion Dynamics and Emotion Regulation

As a psychological phenomenon, emotions can be described in relation to motivational, cognitive and behavioural aspects of personality [6]. Although emotion dynamics is a broad concept with various conceptualisations, studies on this topic usually focus on four affective features, i.e., (i) mean level of positive affect (PA) and negative affect (NA) as well as (ii) variability, (iii) instability and (iv) inertia of PA and NA [7]. Whereas PA and NA mean levels constitute the aggregated intensity of PA and NA, other indicators of affect dynamics, such as variability (generally determined by the within-person standard deviation, SD), instability (measured using the mean squared successive difference, MSSD), or inertia (assessed using the autoregressive slope parameter), take into account fluctuations in emotional states [8]. To calculate emotion dynamics indices, it is necessary to include repeated assessments, at least in the case of the latter three indices. Apart from these most often used indicators of emotion dynamics [9], other indices, including emotion differentiation and emotion-network density (for an overview, see [10]) have been proposed as well. Emotion differentiation (or emotion granularity) is a construct that refers to one’s ability to identify momentary discrete emotional states. As a result, individuals can employ applicable emotion regulation strategies. The poorer the ability to distinguish emotional states, the more ineffective or inappropriate employment of the strategy, which may lead to the development or maintenance of psychopathology [11]. Emotion-network density depends on the strength of temporal connections among specific emotions experienced by a given individual in the past and of some external influences [12]. A denser network means more self-predicting, external-resistant and difficult-to-change emotion system. Another important construct appearing in research on emotional functioning is emotion regulation (ER). ER can be defined as a conscious or unconscious process of determining which emotions we experience, when we experience them, as well as how we experience and express them [13]. ER strategies are multicomponent processes related to changes in emotion dynamics and various emotional response components (such as latency, rise time, magnitude, duration and offset of responses in behavioural, experimental, or physiological domains). Those components are interrelated and change in time as the emotion unfolds [14]. Emotion regulation is determined by three factors: emotional awareness, knowledge of goals and emotion-regulatory strategies. Emotional awareness enables the flexible and conscious use of adaptive ER strategies. Goals refer to emotion-regulation achievements and indicate the increasing or decreasing magnitude or intensity of the emotional experience. Strategies indicate the selection of certain approaches to achieve one’s emotion-regulatory goals. Problematic patterns of the aforementioned components may result from emotion dysregulation, i.e., poor emotional-situational engagement or inadequate emotional response to the situation [15].

### 1.2. Emotion Dynamics and Emotion Regulation in Anorexia Nervosa

Contemporary models of AN recognise the role of abnormal emotional functioning in the development and maintenance of this disorder see, e.g., [16]. Difficulties in emotional functioning in AN include maladaptive ER and emotion expression [17], increased negative affect as a response to aversive stimuli [18] and alexithymia [19]. Among other daily affective experience patterns, NA appears to be critical in the development and maintenance of AN. Studies indicate that higher daily levels of negative affect in AN increase the likelihood of dietary restrictions [20]. NA may also rise after various AN behaviours, such as the loss of control eating (LOC), purging, a combination of LOC/purging, or after a weight check. On the other hand, NA decreases after drinking fluids instead of eating and after physical activity. Several studies focused on NA states and their specific forms (i.e., anxiety or tension) in AN, rumination and disordered eating behaviours (such as dietary restriction, LOC or purging), as well as weighing see, e.g., [21,22,23,24]. Research on emotion dynamics and its constructs, such as intensity, lability, differentiation, and inertia, in AN indicate higher daily NA intensity, as well as lower lability and emotion differentiation than in the case of other eating disorders (bulimia nervosa and binge eating disorder) see, e.g., [11]. These results indicate that individuals diagnosed with AN cannot identify momentary emotional experiences well and have poor access to emotion regulation strategies. Inefficient emotion differentiation may lead to maladaptive behaviours, including more frequent vomiting, use of laxatives, dietary restrictions, and self-weighting regardless of the AN subtype [25]. Heightened affective instability in AN has been shown to precede restrictive behaviours [21] or may lead to self-aggressive, maladaptive behaviours including non-suicidal self-injury (NSSI) [26]. From a theoretical point of view, ANR (anorexia nervosa restricting type) and ANB (anorexia nervosa bulimic type) substantially differ in terms of affective instability. The main reason is that restrictive and bulimic pathologies are supposed to play different roles in emotion regulation. There is abundant evidence that the binge-purge cycle functions as a means of emotion regulation in ANB [27]. In AN, restrictive eating patterns are linked to a narrowing of emotional functioning, flattening of affect, and a lack of outward display of emotions. Both the binge-vomit cycle and restriction are emotion suppression strategies. Binging-vomiting, NSSI and other impulsive behaviours are reported after emotion activation, whereas restriction and compulsive behaviours precede the activation of emotions.

Some authors see, e.g., [16] describe ER as a core deficit and maintaining factor of AN [22,28,29]. Theoretical considerations [16] supported by findings from empirical studies [20] indicate that disturbed adaptive ER is an important contributor to maladaptive ER in the form of dysfunctional behaviours, including food restrictions and compensatory behaviours. Insufficient employment of adaptive ER strategies (such as awareness and acceptance of emotions or effective inhibition thereof) and greater use of maladaptive strategies (i.e., avoidance, rumination, and suppression) are associated with this psychopathology. Emotion dysregulation in AN tends to be associated with the presence of an underlying emotional vulnerability that can result in heightened emotional arousal and may trigger various disordered behaviours [21]. However, the majority of empirical findings on ER in AN are based on cross-sectional studies, which mostly rely on retrospective or trait self-report measures. Therefore, these studies do not allow examining direct associations of dynamic processes, i.e., links between exhibited symptoms and fluctuations in the emotional state of a given individual.

### 1.3. Ecological Momentary Assessment and Clinically Relevant Constructs in Psychiatric Populations

Ecological momentary assessment (EMA), also called experience sampling method (ESM) or ambulatory assessment (AA), is a research method that enables collecting multiple observations of temporal behavioural patterns in one’s natural environment, in particular, aimed to examine the associations between changes in affect and clinically relevant variables in the day-to-day life of a person diagnosed with AN. Research methods can range from the use of paper and pencil to advanced technologies, such as mobile phones, palmtops and web-based applications, which nowadays are used most frequently. There are different sampling strategies in EMA, which is why (i) the devices can be programmed by researchers to trigger prompts according to a schedule (the so-called signal contingent), (ii) participants can create self-reports after the occurrence of a predefined event (the so-called event based-triggered EMA), or (iii) a combination of both strategies can be used (i.e., a combined sampling strategy), whereas the selection of the sampling strategy clearly depends on the study goals [30]. This method allows for circumventing many limitations typical of traditional research approaches, i.e., (i) minimisation of the recall bias associated with retrospective self-reports, thereby enhancing the accuracy of the data; (ii) maximisation of the ecological validity and generalisability by assessing data in the participants’ natural habitat, thus avoiding the artificiality of the laboratory testing environment; and (iii) providing multiple observations per day over a longer period, therefore enabling the examination of dynamic processes within participants [31]. There are several advantages of EMA over traditional self-report or laboratory-based measures, such as the possibility of examining micro-temporal relationships and patterns between variables of interest [32,33], reporting on symptoms close to the participants’ experience, as well as the possibility of following the effectiveness of the treatment [34]. However, up-to-date empirical research utilising EMA in persons diagnosed with AN seems rather insufficient compared to other clinical populations. 

### 1.4. Aims

Although several concepts have been put forward regarding the role of emotion dynamics and ER alterations in the development, course, and maintenance of AN see, e.g., [16,17], there are still not enough conclusions based on longitudinal studies focused on the association between AN symptoms and emotional functioning in the daily-life environment. Because of the above-mentioned benefits of EMA, it has been used in studies focused on emotion-related mechanisms underlying eating disorder behaviors in AN. Therefore, we aimed to (a) systematically review, summarize, and synthesize the available results from EMA studies on emotion dynamics and ER in AN, (b) discuss the main findings, their clinical implications, limitations of the studies reviewed, and future directions, as well as to (c) emphasize the importance of the EMA method in the understanding of emotion dynamics and ER in AN.

## 2. Materials and Methods

### Search Strategy

The MEDLINE, PsycINFO, and Scopus search engines were comprehensively searched for articles published in English in peer-reviewed scientific journals until 31 May 2022, with duplications being automatically removed. The following search algorithm was used: anorex* AND (“emotion*” OR “affect*”) AND (“experience sampling” OR “ESM” OR “ecological momentary” OR “EMA” OR “ambulatory assessment” OR “intensive longitudinal”). Additionally, reference lists of the included articles were hand-searched and papers citing them were searched via Scopus. Figure 1 displays the search process, which was conducted in line with the PRISMA guidelines see [35].

Publications were included in our review if they fulfilled the following criteria: reported findings from original empirical studies on emotion dynamics (i.e., PA/NA, variability, instability, inertia) or ER measured with ESM/EMA/AA; involved clinical samples with AN.

## 3. Results

### 3.1. Study Characteristics

The systematic search yielded 27 articles published between 2005 and 2021 (see Table 1). Three studies and one pilot study (with their results published in 20 papers) were conducted in the USA (17 papers report findings from the same study), one study (with its results published in one paper) was conducted in Canada, three studies (with their results published in four papers) were carried out in Germany, and one study (with its results published in two papers) was carried out in Belgium. In 23 studies (with their results published in 19 papers), patients with AN (restrictive and binge/purging types, full-threshold and sub-threshold) were the only group of participants. In three studies (with their results published in six papers), the two clinical groups involved patients with AN and patients with bulimia nervosa (BN). One study involved three groups of patients (i.e., with AN, BN, and binge eating disorder [BED]) and one study included the following clinical groups: patients with AN, BN, and patients with eating disorders not otherwise specified (EDNOS). Three studies (with their results published in three papers) included HC groups. In three studies (with their results published in four papers) results regarding the restrictive subtype of AN (AN-R) vs. the binge-purging subtype of AN (AN-B) were analyzed. The devices used in almost all studies were palmtops (22 papers) and, to a lesser degree, smartphones (five papers). The assessment period was 14 days in 22 papers, seven days in two papers, five days in one paper, two days in one paper, and one day in one paper. The number of prompts per day varied from six to 17 prompts per day; in more detail, participants received six prompts per day in four studies, nine prompts per day in one study, five prompts per day in one study, 17 prompts per day in one study; in one study, the actual number of daily prompts was not specified (For the whole characteristic of the reviewed studies see Table 1).

### 3.2. Emotion Dynamics

Differences in affective variability (spin and affect lability) and affective states (conceptualized as a two-dimensional space: valence and activation) were expected depending on ED type (AN-R, AN-BP, or BN) in the study by Vansteelandt et al. [27]. The researchers assumed that restriction serves to pre-empt the activation of affects and that binge-purging behavior serves to cope with activated unbearable affects. More affective variability in BN compared to AN-R was expected. Any significant differences between those groups in terms of their mean experienced valence or activation were found. However, the patients with AN-R experienced less spin than patients with BN. Based on their results the authors suggested that restriction may not serve as an emotion suppression strategy that pre-empts the activation of emotion, yet the patients with AN-R may stick more rigidly to the same affect, show less lability or variability in the quality of affect and their emotional spectrum may be narrower. In another study by Vansteelandt et al. [37], the authors aimed to further estimate the differences in the variability and serial autocorrelation components of affective instability (which constitutes a measure of emotional inertia) in AN-R, AN-BP, and BN patients. The results indicated that participants scored significantly higher on valence and felt consequently more positive on weekend days compared to weekdays. There were no significant differences among weekdays between the diagnostic groups in mean valence. Further analyses showed no significant differences in the mean activation of affective states for the three diagnostic groups. Moreover, the results of multiple regression analyses with serial correlation implied that AN-R and AN-BP patients may be characterized by higher emotional inertia compared to patients with BN. 

The results of another study [36] suggest that patients with AN had significantly lower mean levels of emotion identification compared to the HCs, which did not increase over the day (in opposite to HCs’ emotion identification growth ability). Moreover, it was observed that HCs improved in emotion identification over the course of the study, whereas in AN it slightly declined, which may suggest that individuals with AN do not use contextual cues and social feedback to facilitate emotion identification as effectively as HCs.

### 3.3. Emotion Dynamics and Emotion Regulation

#### 3.3.1. Emotion Dynamics and Eating Disorder Symptoms

Corte and Stein [39] confirmed that body-weight self-conception may decrease self-esteem and increase negative mood in patients with AN. AN patients had lower levels of self-esteem when the body weight/shape self-schema was activated compared to HCs. This effect was not found when activating non-weight-related categories of self-description (i.e., health and appearance, psychological traits, work and activities, exercise/athleticism, or social relationships). No significant differences were found in the distribution of self-schemas activated in working memory between the AN and BN groups, and between sub-threshold and full-threshold AN participants (see [39]).

In a preliminary study on anorexic behavior, Engel et al. [22] found that patients showed marked variability in mood—both within and between participants—which may suggest the lability of mood in AN. Moreover, affective lability was moderately associated with self-discrepancy (which reflects the differences between who one believes to be and who one would like to be or who one believes he or she should or ought to be) and stress as well as with restrictive behavior and rituals in individuals with AN. It was also found that reported stressful events were strongly associated with restrictive behavior and rituals. The authors suggest that fluctuations in patients’ mood may precede restrictive behaviors or, alternatively, that mood fluctuations may occur “around” these behaviors.

##### Results from the American Sample of Young Women

Seventeen papers [11,21,22,23,25,26,38,41,43,44,45,46,47,48,49,50,52] report findings from the same study involving an American sample of young women. Generally, the authors focused on investigating indicators of emotion dynamics (i.e., mean emotional intensity, emotional lability, and emotional variability), eating disorders symptoms (i.e., restriction, binge eating, vomiting, exercise, meal skipping, self-weighing, checking thighs, checking joints), and dependencies between these constructs. Findings in these papers were reported in three general ways: for the whole sample, by AN subtypes (i.e., AN-R vs. AN-BP), or by the diagnostic threshold (i.e., full-threshold vs. subthreshold).

(1)Results for the Whole Sample

Results of the statistical analyses of the whole-sample indicated higher lability of NA and anxiety/tension on days with high restriction compared to days with no restriction or with binge eating. Moreover, the analyses indicated higher anxiety/tension lability on days with high vs. low restriction [46]. NA lability, but not anxiety/tension, was uniquely associated with dietary restriction [26,47]. However, no significant difference with regard to overall NA or PA was found between days with high vs. low or no restriction [46]. In addition, guilt was significantly higher before and lower after restrictive eating episodes. In contrast, overall NA and overall PA, fear, joviality, and self-assurance did not differ prior to or following such episodes [45]. To sum up, these findings suggest a significant role of lability (but not intensity) of NA and guilt as its specific dimension in restrictive eating behaviors and do not indicate such a role of PA intensity or lability [45,46]. Apart from that, NA lability was independently associated with binge eating, but not with self-induced vomiting [47]. Interestingly, individuals with AN had similar changes in trajectories of NA before and after binge eating compared to those with BN and BED, which suggests that binge eating episodes in AN may be equally strongly associated with NA in BN and BED [52]. Another study showed that patients with AN had lower NA lability compared to those with BN but not to those with BED [11]. Moreover, AN and BN individuals experienced significantly greater NA intensity than individuals diagnosed with BED, whereas there was no difference between the AN and BN groups [11]. 

Binge eating episodes were also associated with a higher likelihood of subsequent purging relative to the loss of control and non-pathological eating [23]. Furthermore, NA strongly predicted purging following non-pathological eating. Interestingly, approximately one-third of purging episodes occurred in the absence of abnormal eating. The results thus suggest that purging in AN may serve to regulate NA in a similar way as binge eating. Moreover, loss of control and overeating might be important determinants of purging in AN. In addition, significant differences between the trajectories for eating disorders behaviors were found, such as the rates of binge eating, self-induced vomiting, skipping meals, dietary restriction, and daily patterns of anxiety [26].

In a study by Lavender, Utzinger et al. [48], anxiety/tension intensity significantly varied across the time of day; however, no significant links were found between the intensity of overall NA or PA and the time of day. A significant variability across days of the week was reported for the level of all affective variables. Specifically, NA and anxiety/tension were most intense on the weekdays and least intense on the weekends. A reverse pattern was found for PA intensity. There was also significant variability in the frequency of all ED behaviors across hours of the day, with exercise and self-weighing occurring most frequently in earlier hours, meal skipping most frequently at hours typical for breakfast and lunch, and binge eating and vomiting most frequently in later hours. However, no significant variability was identified in ED behaviors across days of the week. Taken together, these findings suggest a significant role of the day of the week in variability in the level of emotions and a similar role of time of day in variability in the frequency of ED behaviors and in the intensity of anxiety/tension.

Berg et al. [38] separately analyzed the data from the AN group (for a description of the EMA datasets, see [55]) and from the BN group (for a description of the EMA datasets, see [56]). Taken together, the latter reported significantly greater NA lability than the former. Post-binge ratings of NA in AN were significantly higher than the pre-binge ratings of NA, which means that NA is higher after binge eating than before (whereas the average NA rating was close to trajectories of NA before and after a binge eating episode).

Findings from a study on negative urgency and daily NA confirmed assumptions about their influence on binge eating frequency [40]. Binge eating episodes occurred more frequently on days with higher levels of NA and negative urgency was positively associated with daily NA. Levels of negative urgency and NA were significantly higher in participants who exhibited binge/purging episodes compared to participants without binge/purging episodes during the assessment period [40]. 

Fitzsimmons-Craft et al. [43] found that the increase of negative affect in AN may be associated with both restrictive and non-restrictive eating behavior, however, only restrictive eating was related to an increased likelihood of subsequent eating disorder behaviors. On the other hand, PA was associated with a decreased probability of restrictive eating behaviors [42]. Furthermore, the restrictive behaviors in AN can serve as a means of reducing negative affect, which might be confirmed by the observed reduction in NA after episodes of excessive exercise and the consumption of fluids [42]. This finding is in line with the results of Fitzsimmons-Craft et al. [43] who observed increased PA after restrictive eating behavior vs. non-restrictive eating behavior; however, PA tended to decrease after eating in general. 

The results of the analysis made by Goldschmidt et al. [23] indicate that NA is a predictor of purging following non-pathological eating and suggests that purging may serve as a source to decrease NA in AN. In another study that included a joint sample of patients with AN and BN [41], those who tended not to engage in self-induced vomiting in the hour following binge eating experienced a greater decrease in guilt than those who engaged in such behavior. However, all participants experienced a decrease in NA and guilt in the hour following a binge-eating episode. Interestingly, participants with AN experienced a lower decrease in guilt in the hour following binge eating than individuals with BN. Taken together, these results support the conclusion that types of behaviors individuals engage in within eating disorders symptoms may be associated with differences in emotion dynamics.

(2)Results by AN Subtype: Restrictive and Binge/Purging AN

The results of the analyses by AN subtype are mostly in line with the results for the whole sample and suggest that patients for whom restrictive behaviors are more frequent, may experience different patterns of emotion dynamics as compared to those who show more behaviors specific to AN-BP. Goldschmidt et al. [44] observed that the most pronounced NA accompanies binge eating and loss of control. Dietary restraint was associated with a lower level of NA but was higher than in solitary eating. Moreover, AN-R patients showed higher levels of solitary eating and lower levels of binge eating and loc episodes as compared to AN-BP participants. These results may support the idea that binge eating and dietary restriction are differently associated with NA and, therefore, may serve diverse functions in patients with different types of AN.

The results of other studies revealed that women with AN-R reported lower overall PA, joviality, and self-assurance prior to restrictive eating episodes (in comparison to after occurrences of restrictive eating episodes) as well as higher self-assurance following them. On the other hand, women with AN-BP reported higher overall PA and self-assurance before compared to after restrictive eating episodes [45]. This might indicate a potentially different role of PA and self-assurance in the mechanism of the onset and course of restrictive eating behavior in AN-R and in AN-BP.

In addition, anxiety/tension level was higher in AN-BP than in AN-R [26]. However, high NA and PA instability and their interaction were related to more frequent weight-loss behaviors regardless of AN subtype and emotional intensity [50]. Taken together, these findings imply a differential role of overall PA intensity, its components (i.e., joviality, self-assuredness, and attentiveness), and anxiety/tension in restrictive eating behaviors depending on AN subtype [45], but emotional lability played a similar role in more frequent weight-loss activities in both AN subtypes [50].

(3)Results by Diagnostic Threshold

Findings from statistical analyses by the diagnostic threshold in the American sample of young women indicate that women with full-threshold AN reported more frequent binge eating and purging, but less frequent checking of thighs and joints than those with EDNOS-AN (i.e., subthreshold AN; [49]). These results suggest that the latter group manifests less pathological eating behavior but is more preoccupied with body size.

#### 3.3.2. Emotion Dynamics and Physical Activity

Physical activity is another ER strategy used by individuals with AN (see [52]). For instance, the authors reported the results of a one-day pilot study in which adolescents with AN and HCs responded to hourly questions on momentary affect while wearing an actigraph to measure physical activity. The authors found that adolescents with AN experienced more aversive tension, more NA, and less PA than HCs. Whereas the level of PA increased after physical activity in both groups, the level of NA decreased only in adolescents with AN. However, this interaction effect did not hold statistical significance when correcting for multiple testing. Thus, to confirm a down-regulation effect of physical activity on NA in AN, further research is needed.

Ma and Kelly [53] followed the trajectories in pride and shame preceding and following physical exercises. They found that for women with AN, pride was higher, and shame was lower immediately after exercise compared to later in the day. However, in the hours following physical activity pride decreased and shame (both body/eating shame and general shame) increased again. Ma and Kelly also found that in the hours before exercise, pride increased, whereas shame did not change significantly. Interestingly, their findings indicate that exercise does not have a long-lasting positive impact on shame and pride experienced by women with AN. The transience of the effect may therefore contribute to the maintenance of exercise in these individuals.

#### 3.3.3. Emotion Dynamics and Rumination

Studies using EMA revealed that rumination about eating and weight is common in everyday life among patients with AN. For example, Seidel et al. [24] compared a group of AN patients with HCs and found that the former spent more time thinking about food, body shape, and weight than the latter. Besides that, the relationship between momentary NA, high tension, and rumination at the same time point was considerably stronger in AN patients than in HCs. Additionally, in both groups, higher weight rumination was related to higher NA on the subsequent prompt. Fürtjes et al. [54] provided a follow-up of these findings. In addition to subjective EMA measures of body weight/figure rumination, food rumination, and affect, the authors used plasma leptin levels as a biological marker of undernutrition. The results of the analyses showed that body weight/figure rumination in AN was linked with affect, but not with leptin level, and persisted even after weight restoration. In contrast, food rumination was less related to affect, correlated with leptin level, and decreased during weight restoration. These findings suggest that body weight/figure rumination is a cognitive-affective aspect of AN, and food-related rumination is its physiological aspect, linked to undernutrition.

#### 3.3.4. Summary

The results of the studies on emotion dynamics and emotion regulation point to relationships between particular components of emotion dynamics and maladaptive emotion regulation strategies as a part of the psychopathology of AN. The empirical data mostly point to associations between NA and PA intensity, lability, and positive emotion differentiation with disordered eating behaviors and weight/figure rumination. Figure 2 presents the schematic representation of the most important associations between emotion dynamics components and specific disordered emotion regulation strategies.

## 4. Discussion

### 4.1. General Discussion

The results of EMA-based studies suggest that persons with AN are characterised by altered emotion dynamics compared to individuals diagnosed with other eating disorders and healthy controls. This is especially manifested in disordered self-reports of this group and pertains to all components of investigated emotion dynamics, that is, mean emotional intensity, emotional instability (liability), emotional variability, emotional inertia and emotion differentiation (granularity). Patients with AN seem to present better emotion differentiation than those with BED or BN [11], but worse than HCs [36]. However, there is no evidence for altered mean emotional intensity or emotional variability in individuals diagnosed with AN-R compared to AN-BP and BN [27] as well as in mean levels of affect (valence and activation). Abnormalities in emotion dynamics in AN appear to depend on the subtype, including higher anxiety or NA lability in AN-BP compared to AN-R [21]. The results of EMA-based studies also point to significant links between overall NA and PA intensity, as well as NA and PA lability with various disordered behaviours, such as dietary restriction, binge eating, purging, weighing oneself and fluid overconsumption to curb the appetite see, e.g., [22,25,40,47,50]. In addition, relevant differences in the associations of emotion dynamics and ER strategies by AN subtypes see, e.g., [46,50] and diagnostic thresholds [49] have been identified. Findings from the study on physical activity in AN and its regulatory effect on specific emotions are not entirely consistent (i.e., pride and shame; [53]). However, this effect appears to be short-lived, which may explain the use of physical activity as a maladaptive ER strategy in AN. Research results on rumination suggest that its short-term effect on NA is more pronounced in patients with AN than in HCs, which justifies a higher momentary level of NA in this group [24]. The findings on rumination also indicate a different nature of its two types in AN: rumination about body weight/figure seems to be a cognitive-affective process, but food-related rumination may reflect a physiological symptom induced by undernutrition [24,54].

The results of studies conducted on the general population highlight the role of various factors in the selection of ER strategies in the day-to-day life, for instance, situational context see, e.g., [56]. However, there is still a need to investigate specific ER strategies in AN in response to situations directly associated with affective instability. Moreover, from a theoretical point of view, it is perfectly reasonable to expect substantial differences between AN-R and AN-BP in terms of affective instability. The main reason is that restrictive and binge/purging pathologies are supposed to play a different role in ER. There is abundant evidence that the binge-purge cycle functions as an element of ER in AN-BP [27]. In AN, restrictive eating patterns are linked with a narrowing of emotional functioning, flattening of affect and lack of outward display of emotion. As such, both the binge-purge cycle and restriction constitute emotion suppression strategies that are utilised at different times. Binging-purging, NSSI and other impulsive behaviours are performed after an emotion has been activated, while restriction and other compulsive behaviours precede the activation of any emotion. Taken together, the findings from the study on both emotion dynamics and its relationships with ER indicate a broad spectrum of difficulties in daily emotional functioning in people with AN.

### 4.2. Clinical Implications

The existing evidence-based therapeutic interventions applicable to patients with AN, such as cognitive remediation therapy [57] or schema therapy [58], comprise elements aimed at targeting emotion dynamics and emotion regulation. Since emotion dynamics and ER in individuals with AN seem to be disordered compared to HCs and change throughout the day, ecological momentary interventions (EMI) appear to be a promising tool of therapeutic approach and can be utilised in patients’ everyday lives as important additional support [59]. Advancements in smartphone technology increased the accessibility of therapeutical applications that can be used for therapeutical purposes [60]. They may be applied to augment in-person therapies (such as integrative cognitive-affective therapy or cognitive behavioural therapy enhanced, [61,62]) by delivering coping strategies at a certain moment, either as requested by the patient or in response to available data. These interventions can be specifically linked to the cue that the participant is experiencing (e.g., NA, body dissatisfaction, interpersonal stressor). Enhanced awareness of cues related to disordered eating behaviour coupled with an increased ability to effectively use adaptive ER strategies when they are most needed will likely contribute to overall improvements in emotion dynamics and ER and thus to better outcomes of AN treatment (for reviews on EMI in EDs, see [33,60,63]).

### 4.3. Limitations

Although the existing EMA-based research provided new insights into the symptomatology experienced by patients with AN in their everyday lives, the studies reviewed have some limitations. The majority used small and almost exclusively female samples and a brief EMA duration (<7 days). Moreover, many studies did not include an HC sample for comparison, and even fewer included a clinical control group. Furthermore, there were methodological differences between EMA protocols, including in the duration of time for answering prompts and the number of EMA prompts per day. Additionally, indicators of ED symptoms were mostly based on subjective data. Hence, several potentially important factors that to some extent limit the generalisability of findings, such as the lack of evidence caused by the low sampling frequency, have been identified. It is important to plan the study sampling protocol in order to track courses of ED behaviours that are characterised by short-term dynamics and may be missed as a result of a low sampling frequency. In fact, it has been empirically shown that the processes associated with ED behaviours in eating disorders are quite fast [64].

### 4.4. Future Directions

Future studies should address the aforementioned limitations, but also test possible differences between patients with AN and HCs with regard to other components of emotion dynamics and ER strategies (including the more adaptive ones) so that possible disordered patterns of individuals diagnosed with AN can be detected. Therefore, components of emotion dynamics, such as emotional interdependency across time, emotional dialecticism (bipolarity), emodiversity and other subtypes of emotional lability, including the probability of acute change or emotional switching (for overviews, see [10,65,66]), as well as cognitive ER strategies, for example cognitive reappraisal or distraction, and behavioural ER strategies, such as social support seeking (for overviews, see [67,68]) should be examined. In addition, patients with AN need to be compared to patients with other mental disorders characterised by maladaptive emotion dynamics and ER, such as borderline personality disorder [69] or major depressive disorder [70], in order to identify disorder-specific and transdiagnostic patterns of these phenomena. Given disordered ER in men with AN see [71], this group should also be involved in further studies. Another important issue is to investigate emotional beliefs and situational context as important factors underlying emotion dynamics and ER see [72,73]. Finally, future studies should be based on current guidelines or recommendations concerning EMA-based research in order to be designed and conducted in accordance with rigorous methodological standards (for guidelines on psychopathology research, see [9]; for recommendations on emotion dynamics and ER research, see [8,74].

## 5. Conclusions

The findings of the reviewed studies point to various disturbed components of emotion dynamics as well as to unique associations of maladaptive ER strategies with specific abnormalities in emotion dynamics in AN.

## Figures and Tables

**Figure 1 ijerph-19-13659-f001:**
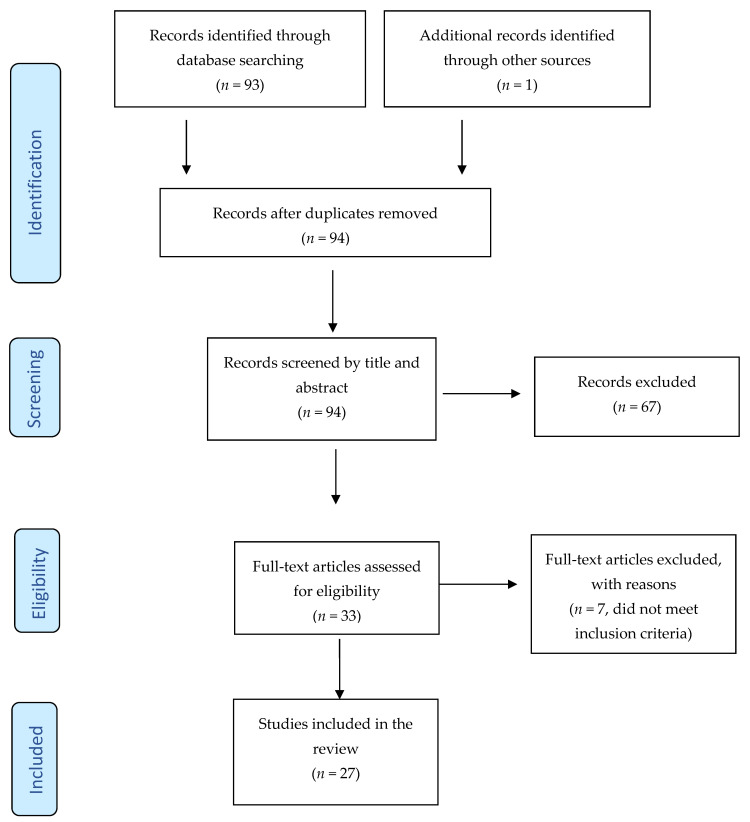
PRISMA flow diagram of the search strategy.

**Figure 2 ijerph-19-13659-f002:**
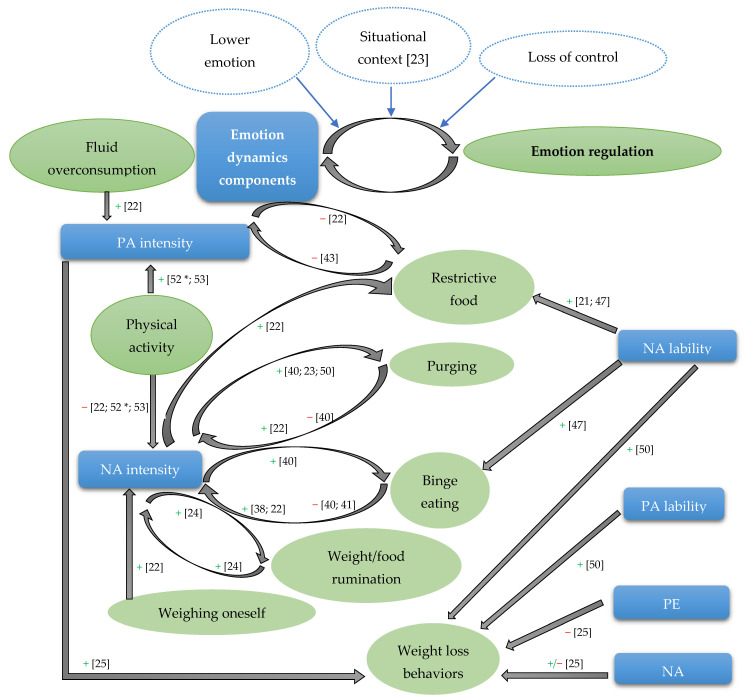
Schematic representation of the relationships of emotion dynamics components and disordered emotion regulation strategies in AN based on empirical data on the significant associations between these constructs. Note. AN—anorexia nervosa; NA—negative affect; PA—positive affect; PE—positive emotion; +—positive association; −—negative association. Weight loss behaviors comprise vomiting, laxative-use, exercising, weighing, checking for fat, and restricting. * The effect did not hold statistical significance when correcting for multiple testing.

**Table 1 ijerph-19-13659-t001:** Characteristics of the reviewed studies.

	Paper	Country	Groups (Sample Size)	Age M (*SD*) [Years]	BMI M (*SD*) [kg/m²]	Constructs	Device/App	Duration	Prompts per Day (by Sampling Scheme)	Compliance Rate	Findings	Limitations
**Emotion Dynamics**	Kolar et al. (2017) [36]	Germany	AN (*n* = 20) HCs (*n* = 20)	AN: 16.0 (1.55) HCs: 15.9 (1.5)	AN: 16.5 (0.9) HCs: n.i.	Momentary aversive tension, emotion identification	Smartphone (AndroidEpiCollect)	2 days	n.i. (every hour excl. individual set night-hours)	n.i.	(1) AN group showed lower emotion identification than HCs; (2) emotion identification improved during the day in HCs; (3) negligible decrease of the emotion identification over time in the AN group	small sample sizes, short EMA duration, no assessment of the quality of valence of emotion, outpatients under treatment
	* Vansteelandt et al. (2013) [27]	Belgium	AN-R (*n* = 21) AN-BP (*n* = 16) BN (*n* = 20)	21.30 (5.64)	AN-R: 14.62 (1.38), AN-BP: 16.02 (1.20), BN: 21.08 (2.59	Differences in affective variability in AN-R and BN, affective states in a two-dimensional space: valence and activation, pulse and spin	palmtop	7 days	9 signal-contingent (random-interval)	n.i.	(1) Diagnostic groups have the same mean levels of affect (valence and activation) but the quality of affective experience spins less in patients with AN-R	Small sample sizes
	* Vansteelandt et al. (2016) [37]	Belgium	AN-R (*n* = 21) AN-BP (*n* = 17) BN (*n* = 20)	n.i.	n.i.	Affective instability (variance and serial dependency)	palmtop	7 days	9 signal-contingent (random-interval)	n.i.	(1) Large part of the total variability is related to between-subject differences (large differences in mean valence between participants); (2) long-term between-day and short-term within-day variances are similar in size, but the largest part of the affective instability is due to the serial autocorrelation within days	Small sample sizes
**Emotion Dynamics and Eating Disorder** **behavior**	§ Berg et al. (2017) [38]	USA	AN (*n* = 118) (including full-threshold and sub-threshold AN participants), BN (*n* = 131)	AN = 25.3 (8.4) BN = 25.34 (7.61)	AN = 17.2 (1.0) BN = 23.92 (5.21)	NA (PANAS), eating disorder behaviors	palmtop	14 days (+2 practice days)	6 signal-contingent (semi-random) + 1 interval-contingent (bedtime) + after ED behavior (event-contingent)	n.i.	(1) Post-binge ratings of NA were significantly higher than the pre-binge ratings of NA for women with AN and BN; (2) the average proximal post-binge ratings of NA were made significantly closer in time to the binge-eating episodes (20 min post-binge) than the average proximal pre-binge ratings of NA (2.5 h pre-binge)	NA was not measured during binge eating episodes so the multilevel model could not describe the trajectory of negative affect during a binge
	† Corte et al. (2005) [39]	USA	AN (*n* = 79; (*n* = 26 AN-R-full-threshold and sub-threshold, *n* = 53) BN—full-threshold and sub-threshold)	21.7 (3.5)	n.i.	Body-weight self-schema, self-esteem, affect, eating disorder behavior (incl. body image, vomiting, laxative use, diuretic use)	palmtop	5 days	5 signal-contingent (semi-random)	AN: 96% BN: 91%	(1) Mean level of self-esteem was lower when body weight/shape was activated in memory compared to when health and (non-weight-related) appearance, psychological traits, work and activities, exercise/athleticism, or social relationships self-schemas were activated in working memory; (2) disordered eating behaviors would be more frequent when body-weight self-schema was activated	n.i.
	† Culbert et al. (2016) [40]	USA	AN (*n* = 82; *n* = 38 full-threshold, *n* = 44 sub-threshold)	25.23 (8.69)	n.i.	Negative urgency (the dispositional tendency to engage in rash action when experiencing negative affect)	palmtop	14 days (+2 practice days)	6 signal-contingent (semi-random) + 1 interval-contingent (bedtime) + after ED behavior (event-contingent)	Signal-contingent: average = 86.84%, median = 91.03%; end-of-day: average = 89.27%, median = 93.54%	(1) Higher levels of negative urgency exhibited a greater frequency of binge eating and purging via a relatively persistent and heightened state of negative emotions	Momentary assessment of NA and binge eating and purging behaviors not methodologically clear (what is the direction of the influence), self-reports about binge/purging episodes, only late-adolescent and adult female group (many in treatment),
	† De Young et al. (2013) [41]	USA	AN (*n* = 47) (including full-threshold and sub-threshold AN participants) BN (*n* = 121)	AN: 25.68 (8.27) BN: 25.21 (7.55)	AN: 16.99 (0.95) BN: 24.00 (5.21)	NA, guilt (scored from NA’s items: dissatisfaction with self, sadness, & anger at self) (PANAS-X); eating disorder behaviors (EDC)	palmtop	14 days (+2 practice days)	6 signal-contingent (semi-random) + 1 interval-contingent (bedtime) + after ED behavior (event-contingent)	n.i.	(1) Decrease in NA and guilt in the hour following binge eating episodes; (2) individuals with AN experienced lower reduction of guilt compared to individuals with BN; (3) individuals who tended not to engage in self-induced vomiting in the hour following binge eating experienced a greater decrease in guilt than those who engaged in such behavior	atypical conceptualization of guilt, self-induced vomiting within an hour after binge eating was investigated only at the individual difference level
	Engel et al. (2005) [42]	USA	AN (*n* = 10) (including full-threshold and sub-threshold AN participants)	27.6 (n.i.)	n.i.	PA, NA (PANAS), eating disorder behaviors	palmtop	14 days (+ several practice days)	6 signal-contingent (semi-random) + 1 interval-contingent (bedtime) + after ED behavior (event-contingent)	random signal compliance 92% (78% within 45 min), 85.7% to end-of-day ratings	(1) Affective lability was correlated with restrictive behavior and rituals	pilot study with a small AN sample only
	§ Engel et al. (2013) [22]	USA	AN *(n* = 118) (including full-threshold and sub-threshold AN participants)	25.3 (8.4)	17.2 (1.0)	NA, PA, eating disorder behaviors	palmtop	14 days (+ several practice days)	6 signal-contingent (semi-random) + 1 interval-contingent (bedtime) + after ED behavior (event-contingent)	87% to signals (77% within 45 min), 89% to end-of-day ratings	(1) Higher daily ratings of NA were associated with a greater likelihood of dietary restriction on subsequent days; NA increased following loss of control, eating, purging, a combination of loss of control and purging, and weighing behavior; (2) NA decreased following the consumption of fluids to curb appetite and exercise; (3) NA increased prior to eating disorders related behaviors and decreased following the occurrence of these behaviors	the precipitants of NA were not measured
	§ Fitzsimmons-Craft et al. (2015) [43]	USA	AN *(n* = 118) (including full-threshold and sub-threshold AN participants)	25.3 (8.4)	17.2 (1.0)	PA, NA (PANAS); restrictive and nonrestrictive eating episodes	palmtop	14 days (+2 practice days)	6 semi-random + 1 interval-contingent (bedtime) + after ED behavior (event-contingent)	87% to signals (77% within 45 min), 89% to end-of-day ratings	(1) NA increased from prebehavior to the time of the behavior but remained stable thereafter for both nonrestrictive and restrictive eating behavior; (2) PA remained stable from prebehavior to the time of the behavior but decreased significantly thereafter; (3) for restrictive eating NA was lower and PA was higher across time than for nonrestrictive eating	the report of restrictive eating episodes was subjective and may not accurately reflect caloric restriction, uneven timing of affect ratings could influence the results
	§ Goldschmidt et al. (2014) [44]	USA	AN (*n* = 118) (including full-threshold and sub-threshold AN participants)	25.3 (8.4)	17.2 (1.0)	NA, guilt, fear, momentary stress (PANAS); eating episodes, eating disorder behaviors (EDC)	palmtop	14 days (+2 practice days)	6 semi-random + 1 interval-contingent (bedtime) + after ED behavior (event-contingent)	87% to signals (77% within 45 min), 89% to end-of-day ratings	(1) Loss of control and binge eating were associated with the highest levels of concurrent NA, and solitary eating with the lowest; (2) restrictive and avoidant eating were associated with equivalent levels of concurrent NA that were lower than those associated with loss of control and binge eating; (3) women with AN-R showed higher levels of solitary eating and lower levels of loss of control and binge eating than women with AN-BP	indicators of eating episodes were based on subjective data
	§ Goldschmidt et al. (2015) [23]	USA	AN (*n* = 118) (including full-threshold and sub-threshold AN participants)	25.3 (8.4)	17.2 (1.0)	NA (PANAS), stressful events, eating disorder behavior (loss of control, overeating, body checking episodes, purging behaviors, eating a high-risk food) (EDC)	palmtop	14 days (+2 practice days)	6 semi-random + 1 interval-contingent (bedtime) + after ED behavior (event-contingent)	n.i.	NA predicted purging following non-pathological eating	indicators of loss of control and binge eating were based on subjective data, temporal relationships among the constructs could not be analyzed
	§ Haynos et al. (2017) [45]	USA	AN (*n* = 118 AN-R: *n* = 73, AN-BP: *n* = 45; *n* = 59 full-threshold, *n* = 59 subthreshold)	25.3 (8.4)	17.2 (1.0)	PA, NA (PANAS), eating episodes and restrictive eating (EDC)	palmtop	14 days (+2 practice days)	6 semi-random + 1 interval-contingent (bedtime) + after ED behavior (event-contingent)	n.i.	(1) Guilt significantly higher before and lower after restrictive eating episodes; (2) overall NA, overall PA, fear, joviality, and self-assurance did not differ prior or following such episodes; (3) women with AN-R: lower overall PA, joviality, and self-assurance before restrictive eating episodes and higher self-assurance after them; (4) women with AN-BP: higher overall PA and self-assurance before restrictive eating episodes	no HCs sample for testing possible differences in emotion dynamics
	§ Haynos et al. (2015) [46]	USA	AN (*n* = 118; *n* = 59 full-threshold, *n* = 59 subthreshold)	25.3 (8.4)	17.2 (1.0)	PA, NA (PANAS), eating episodes, binge eating, and restrictive eating (EDC)	palmtop	14 days (+2 practice days)	6 semi-random + 1 interval-contingent (bedtime) + after ED behavior (event-contingent)	n.i.	(1) Higher lability of NA and anxiety/tension on days with high restriction than on days with no restriction or with binge eating; (2) higher anxiety/tension lability on days with high vs. low restriction; (3) no significant difference in overall NA, anxiety/tension, or PA between days with high vs. low or no restriction	no HCs sample for testing possible differences in emotion dynamics
	§ Lavender, DeYoung, Anestis et al. (2013) [21]	USA	AN (*n* = 116; AN-R: *n* = 71, AN-BP: *n* = 45; *n* = 58 full-threshold, *n* = 58 subthreshold)	25.4 (8.4)	17.2 (1.0)	PA, NA (PANAS), tension/anxiety (POMS), eating disorder behavior (EDC)	palmtop	14 days (+2 practice days)	6 semi-random + 1 interval-contingent (bedtime) + after ED behavior (event-contingent)	87% to signals (77% within 45 min), 89% to end-of-day ratings	(1) NA lability, but not anxiety, uniquely associated with dietary restriction; (2) higher anxiety in AN-BP than in AN-R	no HCs sample for testing possible differences in emotion dynamics
	Lavender et al. (2013) [26]	USA	AN (*n* = 118, including full-threshold *n* = 59 and sub-threshold *n* = 59 AN participants)	25.3 (8.4)	17.2 (1.0)	Daily patterns of anxiety (NA) (PANAS); ED behaviors (binge eating, self-induced vomiting, exercise, body checking, various forms of restrictions) (EDS); trait-level personality pathology variables	palmtop	14 days (+2 practice days)	6 signal-contingent (semi-random) + 1 interval-contingent (bedtime) + after ED behavior (event-contingent)	semi-random signals: 87%, end-of-day ratings: 89%	Overall differences between trajectories for rates of binge eating, self-induced vomiting, body checking, skipping meals, and dietary restriction; distinct daily temporal distributions of ED behaviors across the trajectories in coincidence with high levels of anxiety; traits of personality (affective lability, self-harm, social avoidance, and oppositionality) and the presence of a co-occurring mood disorder were associated with the tendency to experience particular daily anxiety trajectories	Single diagnostic group, only women, small effect size for emotion-related measures and ED, anxiousness results may not be reliable due to influence by the anxious nature of the sample, participant’s self-reported food records
	§ Lavender, Mason et al. (2016) [47]	USA	AN (*n* = 118; *n* = 59 full-threshold, *n* = 59 sub-threshold)	25.3 (8.4)	17.2 (1.0)	PA, NA (PANAS), eating disorder behavior (EDC)	palmtop	14 days (+2 practice days)	6 semi-random + 1 interval-contingent (bedtime) + after ED behavior (event-contingent)	87% to signals (77% within 45 min), 89% to end-of-day ratings	(1) NA lability independently associated with binge eating, but not with self-induced vomiting; (2) NA lability independently associated with dietary restraint	no HCs sample for testing possible differences in emotion dynamics
	§ Lavender, Utzinger et al. (2016) [48]	USA	AN (*n* = 118; *n* = 59 full-threshold, *n* = 59 sub-threshold)	25.3 (8.4)	17.2 (1.0)	PA, NA (PANAS), tension/ anxiety (POMS), eating disorder behavior (EDC)	palmtop	14 days (+2 practice days)	6 semi-random + 1 interval-contingent (bedtime) + after ED behavior (event-contingent)	87% to signals (77% within 45 min), 89% to end-of-day ratings	(1) Significant variability of tension/anxiety intensity across time of day, but no significant links between the intensity of overall NA or PA and time of day; (2) significant variability across days of the week for all affective variables: NA and tension/anxiety were most intense in the middle of the week and least intense on the weekends; reverse pattern for PA intensity; (3) significant variability in all ED behaviors across hours of the day, with exercise and self-weighing most frequent in earlier hours, meal skipping most frequent at hours typical for breakfast and lunch, and binge eating and vomiting most frequent in later hours; (4) no significant variability in ED behaviors across days of the week	no HCs sample for testing possible differences in emotion dynamics
	§ Le Grange et al. (2013) [49]	USA	AN (*n* = 118; *n* = 59 full-threshold, *n* = 59 sub-threshold)	25.3 (8.4) (full-threshold AN: 25.9 (9.1) subthre-shold AN: 24.8 (7.6))	full-threshold AN: 16.6 (1.1) subthre-shold AN: 17.7 (0.7)	PA, NA (PANAS), tension/ anxiety (POMS), eating disorder behavior (EDC)	palmtop	14 days (+2 practice days)	6 semi-random + 1 interval-contingent (bedtime) + after ED behavior (event-contingent)	87% to signals (77% within 45 min), 89% to end-of-day ratings	(1) More frequent binge eating and purging in full-threshold AN than in EDNOS-AN (subthreshold AN); (2) more frequent checking of thighs and joints in subthreshold AN than in full-threshold AN	no HCs sample for testing possible differences in emotion dynamics
	§ Selby et al. (2015) [50]	USA	AN (*n* = 118; AN-R: *n* = 73, AN-BP: *n* = 45)	25.3 (8.4)	17.2 (1.0)	PA, NA (PANAS), tension/anxiety (POMS), eating disorder behavior (EDC)	palmtop	14 days (+ 2 practice days)	6 semi-random + 1 interval-contingent (bedtime) + after ED behavior (event-contingent)	87% to signals (77% within 45 min), 89% to end-of-day ratings	High NA and PA instability and their interaction related to more frequent weight-loss behaviors regardless of AN subtype and emotional intensity	no HCs sample for testing possible differences in emotion dynamics
	§ Selby et al. (2014) [25]	USA	AN (*n* = 118; AN-R: *n* = 73, AN-BP: *n* = 45)	25.3 (8.4)	17.2 (1.0)	PE, NE, PED, NED, weight-loss and evaluation behaviors (EDC)	palmtop	14 days (+ 2 practice days)	6 semirandom + 1 interval-contingent (bedtime) + after ED behavior (event-contingent)	87% to signals (77% within 45 min), 89% to end-of-day ratings	(1) Low PED was associated with more weight-loss behaviors; (2) low PED promoted exercise and high PED promoted weighting; (3) PE and NE intensity positively predicted weight-loss behaviors at the subsequent signal; (4) AN-BP related to more weight-loss behaviors; (5) weight-loss activities predicted elevated levels of PE at the subsequent signal; (6) PED interacted with PE in the prediction of weight-loss activities	no HCs sample for testing possible differences in emotion dynamics
	§ Williams-Kerver et al. (2020) [11]	USA	n = 118 (AN) (including full-threshold and sub-threshold AN participants), n = 133 (BN), *n* = 112 (BED)	AN = 25.3 (8.4) BN = 25.34 (7.61) BED=39.97 (13.37)	AN = 17.2 (1.0) BN = 23.92 (5.21) BED = 35.13 (8.66)	NA (PANAS), intensity, lability, differentiation, inertia	palmtop	14 days (+ 2 practice days; for BED: 1)	6 signal-contingent (semi-random; for BED: 5) + 1 interval-contingent (bedtime) + after ED behavior (event-contingent)	n.i.	AN and BN groups experienced significantly greater NA intensity than the BED group but there was no difference between the AN and BN groups; BN group demonstrated significantly greater NA lability than the AN group but there were no significant differences between the AN and BED or between the BN and BED groups; BN group had significantly higher daily NA differentiation than the AN group and the AN and BN groups demonstrated significantly higher daily NA differentiation than the BED group; BN group had significantly higher scores on inertia compared with the BED group but there were no significant differences between the AN and BN groups or between the AN and BED groups	Not explored whether clinical groups have deficits in daily affective dynamic compared to HCs, not specified relationship in NA in different ED, did not use a standardized set of affect items, methodological limitations among EMA protocol
	§ Wonderlich et al. (2015) [51]	USA	AN (*n* = 118) BN (*n* = 133) Obesity (*n* = 50, (84% females)	25.3 (8.4)	17.2 (1.0)	NA, eating disorder behavior (EDC)	palmtop	14 days (+ several practice days)	6 semirandom + 1 interval-contingent (bedtime) + after ED behavior (event-contingent)	random signal compliance 77%	(1) Moderate to strong concordance between EMA and retrospective measures of negative affect (0.495) and eating disordered behaviors (0.574–0.873)	the timeframe on which the assessments were based did not overlap
**Emotion Dynamics and Physical Activity**	Kolar et al. (2020) [52]	Germany	AN (*n* = 32; AN-R: *n* = 26, AN-BP: *n* = 2, subclinical *n* = 4) HC (*n* = 30)	AN: 16.01 (1.16) HC: 16.36 (2.00)	AN: 16.06 (2.08) HC: 22.22 (2.54)	physical activity measured with an accelerometer, PA, NA, aversive tension	smartphone (movisensXS app), triaxial SOMNOwatchTM accelerometer device	1 day	17 interval-contingent (hourly), possibility of completing questionnaires on request	n.i.	(1) AN group experienced higher levels of aversive tension and NA, as well as lower level of PA than HC; (2) aversive tension was not related to physical activity in the preceding 30 min; (3) NA decreased after physical activity only in the AN group (not significant after correction for multiple testing); (4) PA increased after physical activity in both the AN and the HC group with a stronger effect for AN	Small samples, low adherence to the EMA protocol in AN
	Ma & Kelly (2020) [53]	Canada	AN (*n* = 23)	21.45 (2.99)	17.86 (1.08)	shame, pride, exercise	smartphone (app developed for the study)	14 days	6 semi-random (signal-contingent) + after any episode of exercise (event-contingent)	72.44% (on days when participants reported exercising)	(1) After exercise, pride displayed a decreasing slope, body/eating shame and general shame displayed increasing slopes; (2) before exercise, pride showed an increasing trend	no HCs sample, small AN sample (pilot study)
**Emotion Dynamics and Rumination**	Seidel et al. (2016) [24]	Germany	AN (*n* = 37) HC (*n* = 33)	AN: 16.40 (2.33) HC: 16.51 (3.79)	AN: 14.42 (1.33) HC: 20.63 (1.79)	rumination about food and weight, affect (valence, calmness, energetic arousal)	smartphone (movisensXS app)	14 days	6 semi-random (signal-contingent)	AN *M* = 84.19, *SD* = 11.86 HC *M* = 75.73, *SD* = 12.63	(1) Relations between momentary NA, tension, and rumination were stronger in AN than in HC; (2) Momentary NA was positively associated with a higher amount of disorder-related rumination in patients; (3) rumination about weight led to increased negative affect at the next prompt; (4) rumination about food and weight increased over the course of the study in AN	lower compliance rate in HCs, not all effects could be modeled correctly due to a ceiling effect of rumination and affect in AN
	‡ Fürtjes et al. (2018) [54]	Germany	AN (*n* = 33)	15.4 (1.8)	BMI-SDS *M* = −2.90, *SD* = 0.88	rumination about weight and about food, affect (valence, calmness, energetic arousal)	Smartphone (n.i. about the app)	14 days × 2 (procedure repeated after partial weight recovery)	6 semi-random (signal-contingent)	T1: *M* = 85.28, *SD* = 11.66 T2: *M* = 79.41, *SD* = 16.52	(1) More rumination was related to less positive affect (the effect was stronger for rumination about weight than for rumination about food)	no HCs sample, association between leptin and rumination about food may be based on a shared association with other variables, e.g., hunger

Note. AN—anorexia nervosa; BN—bulimia nervosa; AN-R—anorexia nervosa—restrictive type; AN-BP—anorexia nervosa—binge-purging type; BED—binge eating disorder. NA—negative affect; PA—positive affect; PED—positive emotion differentiation; NED—negative emotion differentiation; PE—positive emotion; NE—negative emotion. EDC—Eating Disorder Checklist; PANAS—Positive and Negative Affect Schedule; POMS—Profile of Mood States; BMI-SDS—standardized BMI. n.i. = no information. * Data come from the same AN sample. § Data come from the same AN sample. † Patients with AN constitute a subset of the AN sample from the study on which papers marked by an asterisk (*) were published. ‡ Data from 26 patients from that sample were included in the statistical analyses reported in Seidel et al. (2016) [24].

## Data Availability

Not applicable.

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
