# Peer review of "Emotion Dynamics and Emotion Regulation in Anorexia Nervosa: A Systematic Review of Ecological Momentary Assessment Studies"

_ijerph, 2022, doi:10.3390/ijerph192013659_

Round 1

Reviewer 1 Report

The manuscript is very well written and organized. It is scientifically sound and logically presented. I just have a few minor comments on it.

1.  In the result section, 3.1 Study Characteristics - the authors stated that three studies (with their results published in 20 papers) were conducted in the USA (line 169) but table 1 shows more than 3 studies in the USA. Did all 20 papers publish different parts of the same study? or is it different studies? If it is the same study then the authors should consider it as one study and draw one conclusion based on that. They should not describe it as paper-wise and draw different conclusions based on what other papers have shown. They should do their independent analysis of each study, draw their conclusions and compare it with the others. It would be great if the authors make it more clear, as currently it is confusing.

2. Line 337, no need of "in" before within.

Reviewer 2 Report

This systematic review aims to summarize and integrate existing findings from experience sampling method (ESM) studies on emotional dynamics and regulation in anorexia nervosa (AN). I appreciate the effort by the authors to summarize the relevant literature in this area, from which the findings are potentially very informative to the understanding of the role of emotional disturbances in the development and maintenance of eating disorders symptoms and shed new light on the treatment options targeting emotion-related processes. I have a few comments for the authors’ consideration, mainly for the Introduction and Discussion.

Introduction: 

-          The concept of “emotional dynamics” needs to be more clearly defined. As “emotional dynamics” is a broad concept, which takes on inconsistent definitions and elaborations (and the underlying components as well) (see Dejonckheere et al., 2019). I suggest including operational definitions (at least for this study) of “emotional dynamics” and its components (e.g. variability, instability, inertia). Therefore, I expect an expansion in section 1.1 to clarify the concept and the underlying components.

Dejonckheere, E., Mestdagh, M., Houben, M., Rutten, I., Sels, L., Kuppens, P., & Tuerlinckx, F. (2019). Complex affect dynamics add limited information to the prediction of psychological well-being. Nature human behaviour, 3(5), 478-491.

 -          Following my previous comment, I think it would be clearer to pin down the exact components of “emotional dynamics” included in this review. Lines 47-48 (p. 2) lists mean, variability, instability and inertia, but somehow emotional differentiation is also included in this review (ref Table 1). Mentioning the exact components of the emotional dynamics to be covered in this review would be helpful to set the scope of the review, and also paving way for the inclusion criteria of the search in this review.

-          Emotional dynamics and emotional regulation are two core elements of this review. However, the rationale for covering both within the same review needs some additional justification and elaboration. The statements “ER strategies are multicomponent processes related to emotion dynamics changes and various emotional response components (such as latency, rise time, magnitude, duration, and offset of responses in behavioral, experimental, or physiological domains). Those components are interrelated and change in time as the emotion unfolds” (lines 50 -54, p. 2) are highly relevant and important here, and further elaboration of the linkage between emotional dynamics and emotional regulation would be informative.

-          The relevance of studying emotional dynamics and emotion regulation in anorexia nervosa should be further elaborated on and explained. For example, are there any theories/ models of eating psychopathology to suggest a role of emotional dynamics (especially for negative affect) and emotional regulation difficulties in the development and maintenance of the symptoms of anorexia nervosa? What kinds of emotional dynamics (e.g. instability and/or intensity) would contribute to the development of these symptoms? How are adaptive and maladaptive emotion regulation strategies and symptoms related?  These perspectives would help readers to appreciate the scientific value of the research question that this review aims to address.

-          The meaning of “attitude” in line 75 (p. 2) is unclear. Please rephrase.

-          Section 1.3 clearly describes the merits of using ESM (in contrast to retrospective self-report) to measure emotional dynamics. The statistical operationalisations of the components of emotional dynamics (e.g. intensity as the mean of the items across the ESM assessments) would be essential here, before Section 1.4 Aims.

Discussion:

-          The discussion could better summarize the wealth of findings in the previous section by having a figure/ schematic representation of the relationships of (specific) emotional dynamics and emotional regulation with symptoms of AN (or subtypes of AN). This would help organize the findings better and offer readers a brief diagrammatic overview/summary of the findings.

-          For Section 4.2 Clinical implications, information on the existing evidence-based psychological/ psychosocial therapies targeting emotion dynamics and emotion regulation, which could supplement the discussion of ecological momentary interventions, would be highly relevant here.

Others:

-          Some slips in the use of English were spotted. E.g.

o   P. 2, line 59: Problematic patterns of above components may be cause[d] by emotion dysregulation

o   P. 6, line 470-471:  Moreover, many studies did not [a verb is missing here] a HCs sample as a comparison group

A more throughout proofreading is recommended.

Round 2

Reviewer 2 Report

I appreciate very much the authors’ effort to address my comments. The manuscript now is very informative and insightful. The new Figure 2 summarize the findings in a concise manner. The synthesis of literature has shed new light on the role of emotional disturbances in the development and maintenance of eating disorders symptoms. Just a minor comment on Figure 2: on the left, some words were overlaid by the arrows.  

Author Response

Dear Reviewer,

Thank you for your comment: we upgraded the appearance of the graph so now the content is better visible.   Kind regards, Magdalena Wayda-Zalewska